# Environmental regulation and corporate tax avoidance—Evidence from China

**Xiaokang Yang**[1], **Junbing Xu**[2], **Minling Zhu**[3]*, **Yinglong Yang**[1]

**1** Institute for Financial & Accounting Studies (IFAS), Xiamen University, Xiamen City, Fujian Province, China, **2** Wang Yanan Institute for Studies in Economics (WISE), Xiamen University, Xiamen City, Fujian Province, China, **3** The School of Economics (SOE), Xiamen University, Xiamen City, Fujian Province, China

* zml_dx@163.com

## Abstract

In this study, we used a difference-in-difference (DID) approach to analyze the effect of environmental regulation on corporate tax avoidance behavior based on China's carbon emissions trading pilot policy of 2013. Our findings were as follows: (1) Environmental regulation has led companies to adopt further tax evasion behaviors. Furthermore, the core conclusion was confirmed after a series of robust and endogenous tests, such as parallel trends and PSM-DID (propensity score matching-difference-in-difference). (2) Environmental regulations increase tax avoidance activities by reducing corporate cash flows. (3) The influence of environmental regulation on firm tax evasion is highly pronounced among non-state-owned enterprises, big-scale enterprises, and enterprises with a high degree of industry competition.

## 1. Introduction

Atmospheric pollution, similar to abnormal climate, melting glaciers, haze worsens, and frequent disasters, has become an important concern worldwide. China is the world's largest emitter of carbon dioxide and sulfur dioxide [1], and is facing increasing environmental pollution and considerable international pressure [2]. In 2009, China formally put forward the carbon intensity binding target, proposing that carbon intensity would decrease by 17% by 2015 compared with that in 2010 [3], and promised to cut the carbon intensity by 40%–45% by 2020; simultaneously, China further proposed to reduce carbon intensity by 60%–65% from 2005 levels by 2030 [4]. In 2013, the National Development and Reform Commission of China authorized Beijing, Tianjin, Shanghai, Chongqing, Guangdong, Hubei, and Shenzhen to execute the carbon emissions trading (CET) pilot policy. In recent years, China's efforts to reduce carbon dioxide emissions have become obvious. In 2017, China achieved the carbon intensity reduction target, and the carbon intensity of China decreased by 46% compared to that of 2005.

The CET pilot policy, as a kind of environmental regulation, has on the one hand, had a significant effect on solving environmental problems in China [5, 6], but on the other hand, we wanted to explore how such a powerful environmental regulation affects micro-enterprises. A review of previous literature indicated that research on the influence of the CET pilot policy is

**Data Availability Statement:** The data underlying the results presented in the study are available from the China Stock Market & Accounting Research Database (CSMAR). The data we used in this study is in the financial statement sub-

database of the company research series in the CSMAR database. If readers want to use the metadata, they can visit the website: https://www.gtarsc.com.

**Funding:** The author(s) received no specific funding for this work.

**Competing interests:** The authors have declared that no competing interests exist.

generally based on regional and industrial perspectives. For example, the European CET policy has effectively reduced the carbon footprint of the Italian manufacturing sector [7]. The Computable General Equilibrium model was used to analyze the carbon footprint intensity of Guangdong Province and it was found that the CET pilot policy can reduce the loss in gross domestic product (GDP) incurred on the premise of achieving the carbon footprint reduction goal [8]. Based on Chinese provincial panel data, it was highlighted that the influence of the CET pilot policy can reduce the carbon footprint intensity by 19%–24% [9]. The significant reduction in the industry's carbon emissions and carbon emission intensity owing to policy implementation was demonstrated using the difference-in-difference (DID) approach [10]. Using PSM-DID, the CET pilot policy was shown to have greatly reduced carbon emissions [11]. The CET pilot policy not only help reduce the regional carbon emission level, but also promote the level of local employment [12]. Owing to the policy, the land supply for energy-intensive industries was reduced by 25% [13]. However, there are few articles on the effect of the CET pilot policy on micro-enterprises. For example, the policy increased pilot enterprises' technological innovation levels [14]. This policy also supported increases in stock returns [15]. The influence of the policy on corporate tax avoidance has, however, not been studied. Theoretically, the CET pilot policy (strong environmental regulation) increases the total cost, production costs, and inventory costs of corporations [16], which decreases the available internal cash. Therefore, corporate managers will engage in additional tax avoidance activities to ensure business operations. In contrast, to comply with strict environmental regulations, companies must install additional pollution control equipment, therefore in the short term, this will also lead to a decrease in the available internal cash, which strengthens the company's motivation for an increase in tax avoidance activities.

Taking the 2013 China carbon emissions trading pilot policy as the background, in this study we utilized the DID approach to analyze the effect of environmental regulations on firm tax avoidance. We found that: (1) Environmental regulation has led companies to adopt further tax evasion behaviors. Furthermore, the core conclusion was confirmed after a series of robust and endogenous tests, such as parallel trends and PSM-DID. (2) Environmental regulations increase tax avoidance activities by reducing corporate cash flows. (3) The influence of environmental regulation on firm tax evasion is obvious among non-state enterprises, small-scale enterprises, and enterprises with a high degree of industry competition. The policy contribution of this article is that when the government implements environmental regulation policies, it should consider the possible impact on micro-enterprises.

Our study makes the following contributions. First, based on the 2013 CET pilot policy as a quasi-natural experiment, our research accurately identifies the causal effect of environmental regulation on firm tax avoidance. Previous research on the influence of environmental regulation on enterprises may be plagued by endogenous factors. In this study, through quasi-natural experiments, we use the DID method to accurately estimate the causal effect of environmental regulation on corporate tax avoidance.

Second, we investigated the effects of the CET pilot policy on micro-enterprises. Most of the previous evaluations of the Chinese CET pilot policy has been based on provincial research [13]. This study assesses the impact of environmental regulation on micro-enterprises from the perspective of enterprise tax evasion.

Finally, this study enriches the literature on factors influencing enterprise tax evasion. Previously, the analysis of the influence factors of enterprise tax evasion has mainly focused on the company-level factors [17, 18] and the external environmental factors of the company [19, 20]; in this study we evaluated enterprise tax evasion from a new perspective, environmental regulation.

The remaining paper is organized as follows: Section 2 provides the institutional background, Section 3 introduces the research design and empirical sampling, Section 4 discusses the basic empirical results, and Section 5 concludes and discusses policy implications.

## 2. Institutional background

Since the "Kyoto Protocol" was signed in October 1997, the completely new notion of reducing greenhouse gas emissions through market mechanisms has been developing continuously. Especially during the period 2002–2005, the UK, Australia, and the EU have successively built carbon emissions trading markets, marking a leap from concept to practice. Carbon emission rights trading is defined as the purchase (sale) of additional (excess) carbon emission rights in the market according to the carbon emission allowances stipulated by the government. The consensus is that the nature of the goods given the carbon emission rights will produce an effect on the production decision of the company. The EU emission system is the largest multinational carbon emissions trading system, covering 24 countries. The regional carbon trading system includes countries and regions such as California, Tokyo, New Zealand, and South Korea.

Recently, China has actively explored the establishment of a domestic carbon emissions trading system to realize a market-based environmental monitoring method for greenhouse gas emission reduction. This is because, on the one hand, driven by the global low-carbon development trend, financial innovation for carbon emission exchanges will become a main driving force of future economic growth; on the other hand, even if the Clean Development Mechanism supplied by China ranks first in the world, the lack of pricing power and low prices have led to a serious hiatus of China's carbon assets. In October 2011, the National Development and Reform Commission of China issued the "Notice on Pilot Work on Carbon Emission Trading Rights" and officially approved Beijing, Tianjin, Shanghai, Chongqing, Hubei, Guangdong, and Shenzhen to implement a CET pilot policy. Although the scale of carbon footprint trading in the experimental region of China is relatively small compared to the scale of EU carbon footprint trading, the pace of development is rapid.

## 3. Research design

### 3.1 Sample selection

In this study, public firms in China were selected as the primary samples. The sample period was from 2010 to 2016. We screened the sample according to the following criteria: (1) excluding the financial industry and special treatment (ST) samples. ST company means the company with an abnormal financial status or other conditions.; (2) excluding samples in which the actual tax burden is less than 0 or greater than 1; (3) excluding companies with only one year of observation; and (4) excluding companies listed after 2009. The data on A-share public companies uses CSMAR data. The CSMAR database is an authoritative financial database developed for experts and scholars from universities, financial securities institutions, and social research institutions to study China's financial economy. To avoid the impact of extreme values on the results, we used a 1% head-to-tail winsorization process on corporate financial indicator data.

### 3.2 Empirical model and variable description

Following previous research [21, 22], the empirical model design of this study is as follows:

$$TA_{i,t} = \alpha_0 + \alpha_1 Treat_i * Post_t + \beta_j \sum Control + \delta_t + \gamma_i + \varepsilon_{i,t} \qquad (1)$$

In model (1), *TA* is a dependent variable for measuring corporate tax avoidance. We use a tax avoidance index expressed as the difference between the actual and nominal tax rates [23]. Therefore, *TA = ETR−TAX*, where *ETR* is the actual income tax rate of the enterprise, using Porcano to divide the income tax expense by pre-tax earnings [24], the difference between the income tax expense and deferred income tax expense divided by the pre-tax earnings. TAX is the enterprise's nominal tax rate. We selected Beijing, Tianjin, Shanghai, Chongqing, Hubei, and Guangdong, which have high levels of environmental regulation [25]. Therefore, the variable *Treat* is equal to 1 if the registered place of an enterprise is in any one of these provinces; otherwise, *Treat* is equal to 0. Because the CET policy was piloted in 2013, if the year was greater than or equal to 2013, *Post* is equal to 1, else, *Post* is equal to 1. The *Control* represents the control variables. Following previous research on tax evasion [23], we selected the following control variables: firm leverage (*Lev*), firm size (*Size*), fixed assets ratio (*NetFi*), intangible assets ratio (*NetIn*), return on assets (*Roa*), firm age (*Age*), the four major accounting firm supervisions (*Foac*), and the nature of the company's property rights (*Soe*). In addition, the model adds a year fixed effect ($\delta_t$) to control the influence of certain nationwide changes that occur in a fixed year on corporate tax avoidance and corporate fixed effect ($\gamma_i$) to control all possible impacts on corporate tax avoidance, whereas not changing corporate characteristics with time. Table A1 in the S1 Appendix presents the explicit circumscription of each variable.

## 4. Empirical analysis

### 4.1 Summary statistics

In Table 1, the mean values of *TA*1 and *TA*2, the two different measured of dependent variable, are both approximately −0.007, which shows that, on average, the actual tax burden is less than the nominal tax burden. Hence, we can see that Chinese listed companies, on average, will engage in tax avoidance activities. The standard deviations are approximately 0.11, which indicates that there are different degrees of tax avoidance between enterprises. Judging from the standard deviations of other explanatory variables, there is a certain degree of difference in the characteristics of enterprises, and their tax avoidance behaviors may be affected by this difference.

### 4.2 Benchmark regression

First, in this study, we verified whether environmental regulation affects tax avoidance. From columns (1) and (2) of Table 2, we found that only the firm with leverage and fixed time

**Table 1. Descriptive statistics.**

| Variable | Obs. | Mean | Sd | Min | Max |
|---|---|---|---|---|---|
| TA1 | 9390 | -0.007 | 0.112 | -0.250 | 0.401 |
| TA2 | 9390 | -0.007 | 0.110 | -0.250 | 0.401 |
| Treat*Post | 9390 | 0.182 | 0.386 | 0 | 1 |
| Lev | 9390 | 0.493 | 0.219 | 0.064 | 1.280 |
| Size | 9390 | 22.170 | 1.401 | 18.550 | 26.070 |
| NetFi | 9390 | 0.235 | 0.181 | 0.001 | 0.757 |
| NetIn | 9390 | 0.049 | 0.062 | 0.000 | 0.390 |
| Roa | 9390 | 0.052 | 0.053 | -0.128 | 0.283 |
| Age | 9390 | 2.783 | 0.342 | 1.386 | 3.332 |
| Foac | 9390 | 0.933 | 0.250 | 0.000 | 1.000 |
| Soe | 9390 | 0.541 | 0.498 | 0.000 | 1.000 |

**Table 2. Environmental regulation and corporate tax avoidance.**

|  | TA1 | TA2 | TA1 | TA2 |
|---|---|---|---|---|
|  | (1) | (2) | (3) | (4) |
| Treat × Post | -0.0136*** | -0.0132*** | -0.0119** | -0.0116** |
|  | (-2.7059) | (-2.6992) | (-2.4389) | (-2.4325) |
| Lev |  |  | -0.0359** | -0.0349** |
|  |  |  | (-2.4451) | (-2.4778) |
| Size |  |  | 0.0214*** | 0.0206*** |
|  |  |  | (4.8725) | (4.9187) |
| NetFi |  |  | -0.0181 | -0.0127 |
|  |  |  | (-0.8218) | (-0.6052) |
| NetIn |  |  | -0.0484 | -0.0439 |
|  |  |  | (-0.8561) | (-0.7757) |
| Roa |  |  | -0.3005*** | -0.2804*** |
|  |  |  | (-7.1047) | (-6.7077) |
| Age |  |  | -0.0413** | -0.0395** |
|  |  |  | (-2.1638) | (-2.1406) |
| Foac |  |  | 0.0380*** | 0.0347*** |
|  |  |  | (3.1680) | (3.1807) |
| Soe |  |  | -0.0031 | 0.0024 |
|  |  |  | (-0.2345) | (0.1978) |
| _Cons | -0.0043*** | -0.0046*** | -0.3584*** | -0.3476*** |
|  | (-4.7372) | (-5.2123) | (-3.4190) | (-3.4368) |
| Firm/Year FE | YES | YES | YES | YES |
| Obs | 9392 | 9392 | 9390 | 9390 |
| Adj_R2 | 0.392 | 0.399 | 0.409 | 0.414 |

[a] T-statistics of clustering to enterprises are shown in parentheses

[b] ***, **, and * indicate significance at the 1%, 5%, and 10% levels, respectively.

effects, the coefficient of *Treat × Post* was significantly negative with *TA*1 or *TA*2 at a confidence level of 1%. After controlling for corporate characteristics, the coefficient of *Treat × Post* was still positive at the 5% confidence level, as shown in columns (3) and (4). Therefore, it was verified that environmental regulation encourages corporate tax avoidance behaviors.

## 4.3 Robustness and endogenous test

To ensure the reliability of the benchmark regression conclusions in this study, the following robust and endogenous tests were performed on the empirical results presented in Table 2.

**4.3.1 Event study.** This section utilizes the event research method to specifically test policy effects. We replaced the dummy variable *Post* in the benchmark DID model with dummy variables *Before* and *After* of each year. Before1 indicates that the year one year before the policy (2012) is equivalent to 1, other years are equivalent to 0, and *After0* indicates that the year of the policy (2013) is 1 and the other years are equivalent to 0; other dummy variables are similarly constructed to obtain an expanded DID model (2).

$$TA_{i,t} = \alpha_0 + \beta_j \sum_{j=1}^{3} Treat * Before_j + \rho_k \sum_{k=0}^{3} Treat * After_k + \lambda_l \sum Control + \delta_t + \gamma_i + \varepsilon_{i,t} \quad (2)$$

From Column (1) and (2) of Table 3, the coefficients of *Treat × Before1*, *Treat × Before2*,

**Table 3. Event study.**

|  | TA1 | TA2 |
|---|---|---|
|  | (1) | (2) |
| Treat × Before3 | -0.0017 | -0.0045 |
|  | (-0.2799) | (-0.7260) |
| Treat × Before2 | -0.0013 | -0.0029 |
|  | (-0.1840) | (-0.4122) |
| Treat × Before1 | -0.0056 | -0.0075 |
|  | (-0.7566) | (-1.0156) |
| Treat × After0 | -0.0191** | -0.0216*** |
|  | (-2.5716) | (-2.9769) |
| Treat × After1 | -0.0148* | -0.0166** |
|  | (-1.8664) | (-2.0999) |
| Treat × After2 | -0.0147* | -0.0155* |
|  | (-1.7448) | (-1.8786) |
| Treat × After3 | -0.0051 | -0.0039 |
|  | (-0.5534) | (-0.4412) |
| Cont_Vars | YES | YES |
| Firm/Year FE | YES | YES |
| Obs | 9390 | 9390 |
| Adj_R2 | 0.409 | 0.414 |

and *Treat × Before3* are seen to be insignificant, indicating that the empirical results support the parallel trend hypothesis, because there is no obvious difference between the experimental group and the control group before the policy. Furthermore, the absolute value of the coefficients from *Treat × After0* to *Treat × After3* gradually decreases, and the significance decreases. Therefore, this shows that the influence of environmental regulation on companies is relatively the greatest in the current period of the policy, and the more the period is in the future, the weaker is the influence of environmental rules on corporate tax evasion.

### 4.3.2 Other robustness and endogeneity tests.

1. Replacing the tax avoidance variables (Rep_TA). In this section, we used $TA3 = ETR3 - TAX$ (income tax expense/[pre-tax profit-deferred income tax expense/nominal rate]) as the tax avoidance and re-regressed model (1). As shown in column (1) of Table 4, there was a significantly negative relationship between the core explanatory variable *Treat × Post* and tax avoidance *TA3* at a 5% confidence level.

2. This section excludes the interference from the tax policies, such as replacing the VAT business tax in 2012 (VAT reform policy) and the 2014 fixed asset depreciation pilot policy (Exc_Tax_Pol). The VAT reform policy has a relatively large impact on the service industry, whereas the fixed asset depreciation pilot policy has an increased impact on the six major industries, such as the biopharmaceutical manufacturing industry in the pilot industry. Therefore, in this section, we re-regressed model (1) by excluding the samples of the seven major industries affected by the policy. As shown in column (2) of Table 4, there was a significantly negative relationship between the core explanatory variable *Treat × Post* and tax avoidance at a 5% confidence level.

3. To exclude the influence of time trends on the main conclusions, in this section we controlled the time trend items (Con_Tim_Trend). In the current study we added time trend

**Table 4. Robustness and endogenous test.**

| Panel A | TA3 | TA1 | | |
|---|---|---|---|---|
| | (1) | (2) | (3) | (4) |
| Treat × Post | -0.0119** | -0.0133** | -0.0094* | -0.0119** |
| | (-1.9923) | (-2.4100) | (-1.8713) | (-2.1075) |
| Rep_TA | YES | | | |
| Exc_Tax_Pol | | YES | | |
| Con_Tim_Trend | | | YES | |
| Exc_Eig_Reg | | | | YES |
| Obs | 9390 | 7577 | 9390 | 6952 |
| Adj_R2 | 0.385 | 0.424 | 0.413 | 0.438 |
| **Panel B** | **TA1** | | | |
| | (5) | (6) | (7) | (8) |
| Treat × Post | -0.0119** | -0.0140*** | -0.0212*** | -0.0127** |
| | (-2.4389) | (-2.5997) | (-3.1368) | (-2.2647) |
| Con_SO2_Pol | YES | | | |
| Sam_2011_2014 | | YES | | |
| PSM-DID | | | YES | |
| Two_DID | | | | YES |
| Cont_Vars | YES | YES | YES | YES |
| Firm/ Year FE | YES | YES | YES | YES |
| Obs | 9390 | 4910 | 3967 | 2982 |
| Adj_R2 | 0.409 | 0.515 | 0.426 | 0.417 |

[a] T-statistics of clustering to enterprises are shown in parentheses

[b] ***, **, and * indicate significance at the 1%, 5%, and 10% levels, respectively.

items and control variables [26]. As shown in column (3) of Table 4, the core explanatory variables remained significant.

4. The influence of the eight central regulations in 2012 (Exc_Eig_Reg) was excluded. As the disclosure of tax evasion activities would result in the loss of managerial reputation, the eight regulations will inhibit the tax evasion activities of state enterprise executives to a certain extent. We controlled for the impact of the eight regulations on tax evasion by constructing the ratio of in-service consumption to operating income [27, 28]. The empirical results are listed in Table 4. As shown in column (4), the core explanatory variables *Treat × Post* and tax avoidance were still significant.

5. During the sample period, China also implemented other environmental regulation policies (Con_SO2_Pol). In May 2007, the State Council issued the "Comprehensive Work Plan for Energy Conservation and Emission Reduction" requiring provinces to raise the standard for the collection of unit sewage charges. When the province implemented the $SO_2$ emission standard policy in a certain year, the variable $SO_2$ was considered equal to 1 in that province, else $SO_2$ was equal to 0. As shown in column (5) of Panel B in Table 4, the core explanatory variables *Treat × Post* and tax avoidance were still significant.

6. To eliminate the interference of policies in other years, we narrowed the sample from 2011 to 2014 (Sam_2011_2014). As shown in column (6) of Panel B in Table 4, the core explanatory variables *Treat × Post* and tax avoidance were still significant.

7. To better alleviate the endogenous problems caused by the self-selection of the sample, this part of the analysis used PSM-DID and selected firm leverage (*Lev*), firm size (*Size*), return on assets (*Roa*), and firm age (*Age*) as the covariates, through the 1:1 neighbor matching method. As shown in column (7) of Panel B in Table 4, the core explanatory variables *Treat* × *Post* and tax avoidance were still significant.

8. We re-estimated model (1) by constructing a two-stage DID model to deal with potential sequence-related problems (Two_DID) [29]. Specifically, we first took 2013 as the time node and divided the sample period into two stages: before corporate environmental regulation (2010–2012) and after corporate environmental regulation (2013–2016). At each stage, the arithmetic mean of each firm's variables was calculated. Through this method, we can effectively compare the average effect of environmental rules on the degree of tax evasion by enterprises. As shown in column (8) of Panel B in Table 4, the estimated coefficient of *Treat* × *Post* was significantly negative, which again shows that the strengthening of environmental regulations significantly increases the degree of tax avoidance by enterprises.

## 4.4 Mechanism analysis: Company's cash flow

Strict environmental control not only increases the total cost, production cost, and inventory cost (Berman and bui, 2001), but also makes enterprises bear additional expenses owing to the installation of more pollution purification equipment. In the short term, it will cause the company's available cash or cash equivalents to decrease, thereby prompting companies to engage in tax evasion activities to increase ready money. We uses *Cash*, the cash flow variable (the ratio of cash and cash equivalents to assets) as the explanatory variable. As shown in column (1) of Table 5, the core explanatory variables *Treat* × *Post* and *Cash* are significantly negative. In other words, environmental regulation will lead to a decrease in the cash held by the company, which pushes companies to engage in more tax evasion to ensure the company's cash flow.

## 4.5 Heterogeneity analysis

### 4.5.1 Heterogeneity analysis: Ownership structure.
According to the attribute of the actual controller of the enterprise, the samples were divided into state and non-state companies. We regressed the SOE and non-SOE samples. As shown in columns (1) and (2) of Table 6, environmental regulation has a strong and highly significant influence on tax evasion in non-SOE samples. The reason is that state enterprises have the advantages of "political

**Table 5. Mechanism analysis.**

|                          | Cash        |
|--------------------------|-------------|
|                          | (1)         |
| Treat × Post             | -0.0116**   |
|                          | (-2.1093)   |
| Cont_Vars                | YES         |
| Firm/Year FE             | YES         |
| Obs                      | 9390        |
| Adj_R2                   | 0.643       |

[a] T-statistics of clustering to enterprises are shown in parentheses
[b] ***, **, and * indicate significance at the 1%, 5%, and 10% levels, respectively.

**Table 6. Heterogeneity analysis: Ownership structure.**

| | TA1 | |
|---|---|---|
| | **NSOE** | **SOE** |
| | **(1)** | **(2)** |
| Treat × Post | -0.0168** | -0.0133** |
| | (-2.2703) | (-2.0186) |
| Cont_Vars | YES | YES |
| Firm/ Year FE | YES | YES |
| Obs | 4283 | 5049 |
| Adj_R2 | 0.401 | 0.443 |

[a] T-statistics of clustering to enterprises are shown in parentheses

[b] ***, **, and * indicate significance at the 1%, 5%, and 10% levels, respectively.

connection" and "tax and fee concessions." When faced with the decline in available cash flow caused by environmental regulations, state companies can obtain funds from government or banks relatively easily, and non-SOEs may commence tax evasion to increase cash to make up for the impact of the decline in cash flow due to environmental regulations.

**4.5.2 Heterogeneity analysis: Firm size.** Enterprises can be divided into large-scale and small-scale companies according to the size of the enterprise assets. Large-scale and small-scale companies were regressed separately. As shown in columns (1) and (2) of Table 7, the influence of environmental regulations on corporate tax evasion was more significant in large-scale company samples than in small scale companies. This is because Chinese environmental regulations are generally government-led, and local governments have corresponding environmental indicators and tasks. Generally, the emission and pollution levels of large-scale enterprises are relatively high. Therefore, the environmental supervision of large-scale enterprises can quickly reach the environmental indicators and tasks required by higher-level governments. Therefore, the department of environmental regulations has encouraged large-scale enterprises to accept strong environmental regulations. Simultaneously, to maintain a stable cash flow, enterprises have strong incentives to avoid tax.

**4.5.3 Heterogeneity analysis: Industry competition.** According to the Herfindahl index of asset size, companies can be divided into companies with large industry competition and small industry competition. As shown in columns (1) and (2) of Table 8, the influence of environmental regulations on corporate tax evasion is extremely significant among companies

**Table 7. Heterogeneity analysis: Firm size.**

| | TA1 | |
|---|---|---|
| | **Small** | **Big** |
| | **(1)** | **(2)** |
| Treat × Post | -0.0023 | -0.0229*** |
| | (-0.3664) | (-2.9036) |
| Cont_Vars | YES | YES |
| Firm/Year FE | YES | YES |
| Obs | 4592 | 4598 |
| Adj_R2 | 0.437 | 0.434 |

[a] T-statistics of clustering to enterprises are shown in parentheses

[b] ***, **, and * indicate significance at the 1%, 5%, and 10% levels, respectively.

**Table 8. Heterogeneity analysis: Industry competition.**

|  | TA1 | |
| --- | --- | --- |
|  | **High** | **Low** |
|  | **(1)** | **(2)** |
| Treat × Post | -0.0143** | -0.0107 |
|  | (-1.9888) | (-1.5192) |
| Cont_Vars | YES | YES |
| Firm/Year FE | YES | YES |
| Obs | 4433 | 4957 |
| Adj_R2 | 0.404 | 0.414 |

[a] T-statistics of clustering to enterprises are shown in parentheses

[b] ***, **, and * indicate significance at the 1%, 5%, and 10% levels, respectively.

with high levels of industry competition. This is because when a company's cash flow declines owing to environmental regulations, companies with a high degree of competition in the industry are concerned about the decline in their competitiveness; thus, their motivation for tax avoidance is strengthened. Owing to space limitations, the heterogeneity analysis uses only *TA*1 as the core explanatory variable. The regression results of *TA*2 are presented in the S1 Appendix.

## 5. Conclusions and policy implications

### 5.1 Conclusions

Based on data on A-share listed companies from 2010 to 2016, the influence of environmental rules on tax evasion by enterprises was evaluated using the DID method. We found that (1) environmental regulation has led companies to adopt more tax evasion behaviors. Furthermore, the core conclusion was confirmed after a series of robust and endogenous tests, such as parallel trends and PSM-DID. (2) Environmental regulations increase tax avoidance activities by reducing corporate cash flows. (3) The effect of environmental rules on corporate tax evasion is highly obvious among non-state enterprises, big-scale enterprises, and enterprises with a high degree of industry competition.

### 5.2 Policy implications

This study provides policymakers with two policy suggestions. First, it is suggested that the government should consider the effect of environmental rules on corporate tax avoidance behavior when designing future environmental policy, and more generally, the distorted effect of environmental rules on corporate tax evasion behavior. Second, it is recommended that the government adopt different environmental regulations for different enterprises. Because there are different types of companies in the market, and their respective companies have corresponding characteristics, a one-size-fits-all environmental supervision policy may affect individual companies and cause them to be unable to operate normally.

## Supporting information

**S1 Appendix.**
(DOCX)

## Author Contributions

**Conceptualization:** Xiaokang Yang.

**Data curation:** Junbing Xu, Yinglong Yang.

**Formal analysis:** Junbing Xu.

**Investigation:** Junbing Xu.

**Methodology:** Junbing Xu.

**Project administration:** Junbing Xu.

**Resources:** Junbing Xu, Minling Zhu.

**Software:** Junbing Xu.

**Supervision:** Junbing Xu.

**Validation:** Junbing Xu, Minling Zhu.

**Writing – review & editing:** Minling Zhu.

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
