## [Decision Letter · Decision Letter 0]

2 Nov 2021

PONE-D-21-25235Environmental Regulation and Corporate Tax Avoidance——Evidence from ChinaPLOS ONE

Dear Dr. Xu,

Thank you for submitting your manuscript to PLOS ONE. After careful consideration, we feel that it has merit but does not fully meet PLOS ONE’s publication criteria as it currently stands. Therefore, we invite you to submit a revised version of the manuscript that addresses the points raised during the review process.

We look forward to receiving your revised manuscript.

Kind regards,

Ricky Chia Chee Jiun

Academic Editor

PLOS ONE

Journal Requirements:

Whilst you may use any professional scientific editing service of your choice, PLOS has partnered with both American Journal Experts (AJE) and Editage to provide discounted services to PLOS authors. Both organizations have experience helping authors meet PLOS guidelines and can provide language editing, translation, manuscript formatting, and figure formatting to ensure your manuscript meets our submission guidelines. To take advantage of our partnership with AJE, visit the AJE website (http://aje.com/go/plos) for a 15% discount off AJE services. To take advantage of our partnership with Editage, visit the Editage website (www.editage.com) and enter referral code PLOSEDIT for a 15% discount off Editage services.  If the PLOS editorial team finds any language issues in text that either AJE or Editage has edited, the service provider will re-edit the text for free.

A clean copy of the edited manuscript (uploaded as the new *manuscript* file)”"

Reviewers' comments:

Reviewer's Responses to Questions

**Comments to the Author**

1. Is the manuscript technically sound, and do the data support the conclusions?

Reviewer #1: Yes

2. Has the statistical analysis been performed appropriately and rigorously? 

Reviewer #1: Yes

3. Have the authors made all data underlying the findings in their manuscript fully available?

Reviewer #1: Yes

4. Is the manuscript presented in an intelligible fashion and written in standard English?

Reviewer #1: Yes

5. Review Comments to the Author

Reviewer #1: COMMENTS

LINE COMMENTS

50 Grammatical error: Revise the statement by deleting the word “can”.

59 Concord error: Check and correct the use of the word “utilize”, use utilizes instead.

68 Incomplete sentence: endogenous …, endogenous problems?

77-80 Tautology: revise

101 Grammatical error: Public “firm”, use firms

121 Grammatical error: “every variable or all variables”

150 Omission: the word “column” was omitted

169 Unclear statement: revise the statement

170 Repetition or words: “time trend items”

186 Check “ROA”: not price-Return on asset

255 Contradiction: line 255 contradicts line 277 and Table 7 on line 234

Check your chapter numbering.

6. PLOS authors have the option to publish the peer review history of their article (what does this mean?). If published, this will include your full peer review and any attached files.

Reviewer #1: **Yes: **Samuel Evergreen Adjavon

---

## [Author Response · Author response to Decision Letter 0]

16 Nov 2021

General response to the editor

Dear editor:

We would like to thank you and the anonymous reviewer for the time spent on our manuscript. The reviews have been incredibly helpful, and our manuscript has further improved because of your suggestions. We have carefully checked each of the reviewers’ points and revised accordingly. The main modifications are marked in red in the revised manuscript. 

Again, we thank you for your review and the opportunity to resubmit the revised version. 

Kind regards,

 Minling Zhu 

Responses to Reviewer’s Comments

First of all, we want to explain that we have sorted out the text and language of the full text according to the requirements of the editor to avoid possible problems such as ambiguity and grammatical errors. At the same time, in order to ensure the professionalism and readability of the article, we invited a professional article editing company-Editage, which is cooperated with Plos One, to modify and polish the article.

1.50 Grammatical error: Revise the statement by deleting the word “can”

⇒Response:

We thank the reviewer for carefully reviewing our manuscript, and there is indeed incorrect to use “can” here. So, we deleted the world “can” here. It should be noted that since the text contains traces of modification, the number of lines corresponding to the text is different from the original paper. Details of the modification is list in the revised paper, and we just provide a brief exhibition of the modification here.

The CET pilot policy not only help reduce the regional carbon emission level, but also promote the level of local employment [12]. Owing to the policy, the land supply for energy-intensive industries was reduced by 25% [13].

2.59 Concord error: Check and correct the use of the word “utilize”, use utilize instead

⇒Response:

We thank the reviewer for carefully reviewing our manuscript, and there is indeed incorrect to use “utilize” here, because paper or study is a third-person singular. So, we use utilizes instead here. Details of the modification is list in the revised paper, and we just provide a brief exhibition of the modification here.

Taking the 2013 China carbon emissions trading pilot policy as the background, in this study we utilized the DID approach to analyze the effect of environmental regulations on firm tax avoidance.

3.68 incomplete sentence: endogenous…, endogenous problems?

⇒Response:

We thank the reviewer for carefully reviewing our manuscript. For line 68, in this sentence, we want to explain that the previous research did not use quasi-natural experiments to identify causality. Therefore, the research conclusions they obtained may not indicate that there is a causal effect among the research objects, that is, endogenous factors will affect the research conclusions. Therefore, we use the term endogenous factors here instead of the term endogenous. Details of the modification is list in the revised paper, and we just provide a brief exhibition of the modification here.

Our study makes the following contributions. First, based on the 2013 CET pilot policy as a quasi-natural experiment, our research accurately identifies the causal effect of environmental regulation on firm tax avoidance. Previous research on the in-fluence of environmental regulation on enterprises may be plagued by endogenous factors.

4.77-80 Tautology: revise

⇒Response:

We thank the reviewer for carefully reviewing our manuscript. In 77-80, this paper indeed exist tautology. So, we can those sentence and revised as follow.

The remaining paper is organized as follows: Section 2 provides the institutional background, Section 3 introduces the re-search design and empirical sampling, Section 4 discusses the basic empirical results, and Section 5 concludes and discusses policy implications.

5.101 Grammatical error:Public “firm”, use firms

⇒Response:

We thank the reviewer for carefully reviewing our manuscript, and there is indeed incorrect to use “firm” here, because public firm is more than one. So, there has to be revised to firms. So, we can those sentence and revised as follow.

In this study, public firms in China were selected as the primary samples. The sample period was from 2010 to 2016.

6.121 Grammatical error: “every variable or all variables”

⇒Response:

We thank the reviewer for carefully reviewing our manuscript. Actually, the Table A1 in the Appendix offers explicit circumscription of each variable, because each variable has its own meaning. So, we use each variable instead every variable.

Following previous research on tax evasion [23], we selected the following control variables: firm leverage (Lev), firm size (Size), fixed assets ratio (NetFi), intangible assets ratio (NetIn), return on assets (Roa), firm age (Age), the four major accounting firm supervisions (Foac), and the nature of the company’s property rights (SoE). In addition, the model adds a year fixed effect (δt) to control the influence of certain nationwide changes that occur in a fixed year on corporate tax avoidance and corporate fixed effect (γi) to control all possible impacts on corporate tax avoidance, whereas not changing corporate characteristics with time. Table A1 in the Appendix presents the explicit circumscription of each variable.

7.150Omission: the world “column” was omitted

⇒Response:

We thank the reviewer for carefully reviewing our manuscript. After checking the text, we find that we actually miss the world “column”, so we add the world “column” here.

From Column (1) and (2) of Table 3, the coefficients of Treat × Before1, Treat × Before2, and Treat × Before3 are seen to be insignificant, indicating that the empirical results support the parallel trend hypothesis, because there is no obvious difference between the experimental group and the control group before the policy.

8.169Unclear statement: revise the statement

⇒Response:

We thank the reviewer for carefully reviewing our manuscript. Actually here we want to express that we have controlled time trend. In order to make is more clearly, we change this sentence as follow.

To exclude the influence of time trends on the main conclusions, in this section we controlled the time trend items (Con_Tim_Trend). In the current study we added time trend items and control variables [26]. As shown in column (3) of Table 4, the core explanatory variables remained significant.

9.170 Repetition or words: “time trend items”

⇒Response:

We thank the reviewer for carefully reviewing our manuscript. Because 170 is near the line 169. So, we revise this paragraph. We revised it as follow, which is the same as the response in 8.

To exclude the influence of time trends on the main conclusions, in this section we controlled the time trend items (Con_Tim_Trend). In the current study we added time trend items and control variables [26]. As shown in column (3) of Table 4, the core explanatory variables remained significant.

10.186 Check “ROA”: not price-Return on asset

⇒Response:

We thank the reviewer for carefully reviewing our manuscript. We checked that ROA, also called the return on assets, is an index used to measure how much net profit is created per unit of assets. So, we change the price return on assets as “return on assets”, which is brief and can convey the meaning. Details of the modification is list in the revised paper, and we just provide a brief exhibition of the modification here.

(7) To better alleviate the endogenous problems caused by the self-selection of the sample, this part of the analysis used PSM-DID and selected firm leverage (Lev), firm size (Size), return on assets (Roa), and firm age (Age) as the covariates, through the 1:1 neighbor matching method. As shown in column (7) of Panel B in Table 4, the core explanatory variables Treat × Post and tax avoidance were still significant.

11.255Contradiction: line 255 contradicts line 277 and Table 7 on line 234

⇒Response:

We thank the reviewer for carefully reviewing our manuscript. Based on the reviewers’ comments, we checked the original text and compared the original text with the regression results. We found that this problem was caused by a clerical error in the process of writing the article, that is, the experimental results were not wrong, but we written wrong in the conclusion. Environmental regulations are more likely to trigger tax avoidance by Big businesses. Details of the modification is list in the revised paper, and we just provide a brief exhibition of the modification of conclusion and Abstract here.

Based on data on A-share listed companies from 2010 to 2016, the influence of environmental rules on tax evasion by en-terprises was evaluated using the DID method. We found that (1) environmental regulation has led companies to adopt more tax evasion behaviors. Furthermore, the core conclusion was confirmed after a series of robust and endogenous tests, such as paral-lel trends and PSM-DID. (2) Environmental regulations increase tax avoidance activities by reducing corporate cash flows. (3) The effect of environmental rules on corporate tax evasion is highly obvious among non-state enterprises, big-scale enterprises, and enterprises with a high degree of industry competition.

In this study, we used a difference-in-difference (DID) approach to analyze the effect of environmental regulation on cor-porate tax avoidance behavior based on China’s carbon emissions trading pilot policy of 2013. Our findings were as follows: (1) Environmental regulation has led companies to adopt further tax evasion behaviors. Furthermore, the core conclusion was con-firmed after a series of robust and endogenous tests, such as parallel trends and PSM-DID (propensity score match-ing-difference-in-difference). (2) Environmental regulations increase tax avoidance activities by reducing corporate cash flows. (3) The influence of environmental regulation on firm tax evasion is highly pronounced among non-state-owned enterprises, big-scale enterprises, and enterprises with a high degree of industry competition.

13.Check your chapter numbering

⇒Response:

We thank the reviewer for carefully reviewing our manuscript. We checked the chapter number and revised 4.4.4 to 4.5.3.

14. according to the editors’ advices, we check the whole text and we add footnote for ST sample and CSMAR data set. We read through the full text and revised the content that was vague or inconsistent in semantics. Based on this, we also asked the professional retouching company Editage, which cooperated with Plos One, to polish it. All changes to the article are presented in the document Revised Manuscript with Track Changes.

Thank you again for this opportunity to revise and resubmit our manuscript. We appreciate it and hope to hear from you soon.

Yours sincerely

---

## [Editor Report · Decision Letter 1]

23 Nov 2021

Environmental Regulation and Corporate Tax Avoidance——Evidence from China

PONE-D-21-25235R1

Dear Dr. Minling Zhu,

We’re pleased to inform you that your manuscript has been judged scientifically suitable for publication and will be formally accepted for publication once it meets all outstanding technical requirements.

Kind regards,

Ricky Chia Chee Jiun

Academic Editor

PLOS ONE
---

## [Editor Report · Acceptance letter]

5 Jan 2022

PONE-D-21-25235R1 

Environmental regulation and corporate tax avoidance - Evidence from China 

Dear Dr. Zhu:

I'm pleased to inform you that your manuscript has been deemed suitable for publication in PLOS ONE. Congratulations! Your manuscript is now with our production department. 

Kind regards, 

on behalf of

Dr. Ricky Chia Chee Jiun 

Academic Editor

PLOS ONE